# Multi-domain automated patterning of DNA-functionalized hydrogels

**Moshe Rubanov**[1], **Joshua Cole**[1], **Heon-Joon Lee**[2], **Leandro G. Soto Cordova**[1], **Zachary Chen**[1], **Elia Gonzalez**[1], **Rebecca Schulman**[1,3,4]*

1 Department of Chemical and Biomolecular Engineering, Whiting School of Engineering, Johns Hopkins University, Baltimore, Maryland, United States of America, 2 Department of Biomedical Engineering, Whiting School of Engineering and the School of Medicine, Johns Hopkins University, Baltimore, Maryland, United States of America, 3 Department of Computer Science, Whiting School of Engineering, Johns Hopkins University, Baltimore, Maryland, United States of America, 4 Department of Chemistry, Krieger School of Arts and Sciences, Johns Hopkins University, Baltimore, Maryland, United States of America

* rschulm3@jhu.edu

**Data Availability Statement:** The raw data is found on the Github associated with the paper: https://github.com/MishaRubanov/MAPDH The processed

## Abstract

DNA-functionalized hydrogels are capable of sensing oligonucleotides, proteins, and small molecules, and specific DNA sequences sensed in the hydrogels' environment can induce changes in these hydrogels' shape and fluorescence. Fabricating DNA-functionalized hydrogel architectures with multiple domains could make it possible to sense multiple molecules and undergo more complicated macroscopic changes, such as changing fluorescence or changing the shapes of regions of the hydrogel architecture. However, automatically fabricating multi-domain DNA-functionalized hydrogel architectures, capable of enabling the construction of hydrogel architectures with tens to hundreds of different domains, presents a significant challenge. We describe a platform for fabricating multi-domain DNA-functionalized hydrogels automatically at the micron scale, where reaction and diffusion processes can be coupled to program material behavior. Using this platform, the hydrogels' material properties, such as shape and fluorescence, can be programmed, and the fabricated hydrogels can sense their environment. DNA-functionalized hydrogel architectures with domain sizes as small as 10 microns and with up to 4 different types of domains can be automatically fabricated using ink volumes as low as 50 μL. We also demonstrate that hydrogels fabricated using this platform exhibit responses similar to those of DNA-functionalized hydrogels fabricated using other methods by demonstrating that DNA sequences can hybridize within them and that they can undergo DNA sequence-induced shape change.

## Introduction

DNA is a versatile tool for performing chemical information processing and storage, and for executing temporal and spatial chemical programs [1, 2]. DNA can be used to write molecular programs, *i.e.*, sets of reaction networks that together process chemical information, such as the concentrations or types of chemicals present in a solution. Molecular programs can have multiple inputs and outputs [3]. Molecular programs can also be localized, i.e., specific

data is found in the main text and supporting
information (SI).

**Funding:** This work was supported by ARO
W911NF2010057, NIH R21CA251027-01A1 and
DARPA HR00112110006.

**Competing interests:** The authors have declared
that no competing interests exist.

molecules can be anchored in place and interact with diffusing molecules to produce spatio-temporal molecular programs. Spatiotemporal molecular programs can sequentially release DNA at prescribed times and locations, and they can generate stable chemical gradients within a microfluidic chamber [4–6]. The DNA molecules that specify a spatiotemporal program can be conjugated to substrates such as hydrogels, surfaces, colloids, cell surfaces, or proteino-somes [3, 7–11]. In such systems, spatiotemporal molecular programs can swell materials, cause them to assemble into 2 and 3-dimensional shapes, or alter their fluorescence or porosity [3, 12–14]. For example, the DNA attached to DNA-functionalized colloids can direct how these colloids self-assemble into specific nanostructures [11, 15] and DNA-embedded protei-nosomes can communicate with their neighbors [9]. Genes can be expressed locally on DNA-functionalized surfaces [16] and DNA-embedded hydrogels can transform input patterns of light into spatial patterns of DNA species [17].

DNA within DNA-functionalized hydrogels can react with free DNA, induce programma-ble shape change, and can be transcribed and translated for cell-free protein synthesis [14, 18–20]. DNA-functionalized hydrogels can likewise act as sensors and transducers to enable the translation of sensory information in the environment to a change in the hydrogel [21]. For example, a DNA-functionalized hydrogel-based sensor can simultaneously detect and remove mercury from water [22] and sensors for DNA-based hydrogels exist for inputs such as miRNA, pH, ATP, temperature, as well as various other stimuli [23]. Multidomain hydrogel architectures can likewise be used to create shape-changing materials with multi-stimulus con-trol, or as sensors capable of integrating multiple stimuli [21, 24]. Here we sought to develop a platform for automatically fabricating multi-domain DNA-functionalized hydrogels with low reagent volumes (100 μL) and high resolutions (down to 10 μm). This platform would enable the fabrication of hydrogels that could compartmentalize DNA within a larger architecture, leading to greater functionality for these hydrogel architectures as shape-changing materials or sensors.

Many different methods for fabricating multi-domain hydrogels have been developed [25]. Extrusion-based hydrogel printing has been used to print 3D multi-domain hydrogels down to 100s of microns [26]. For higher resolutions (10s of microns), light-based methods for fabri-cating hydrogels are generally used. One light-based method for fabricating hydrogels involves digital light processing (DLP), where a digital micromirror device is used to direct light to photopolymerize hydrogels at different locations. Hydrogels fabricated with DLP can have res-olutions down to 7 microns [27]. However, scaling the number of inks within DLP systems remains challenging, as the hydrogels are generally fabricated from within an open vat, and exchanges of inks within a vat are difficult. There exist a few examples in literature where vat exchange enables the automated patterning of multi-domain 3D hydrogels [28, 29]. However, to scale this method towards 10s or even 100s of inks, having a vat for each ink would be costly and impractical. Another example is to exchange inks within a closed vat using a flow control-ler [30]. This method, however, requires milliliters of each ink. We aim to adapt high-resolu-tion DLP-based photopatterning towards a hydrogel fabrication platform that has a scalable number of inks while keeping ink volumes down to 100s of microliters.

Here we develop a platform termed Multi-domain, Automated Photopatterning of DNA-functionalized Hydrogels (MAPDH) which enables the automatic fabrication of multi-domain DNA-functionalized hydrogels within a microfluidic chamber using low ink volumes (down to 100 μL) with an integrated Python script. We characterize the reproducibility of fabricating DNA-functionalized hydrogels between each patterning round (injecting new ink) by measur-ing average fluorescence and hydrogel length. We then demonstrate the ability to fabricate a four-color pixel-art image by photopatterning four different DNA-functionalized hydrogels in specified locations (i.e., pixels) extracted from the image. To test whether DNA anchored

within fabricated hydrogels can react with free DNA, we fabricate another four-domain hydrogel architecture, where two of the four DNA-functionalized hydrogels, anchored with fluorescent DNA, can hybridize with a quencher-modified DNA strand that is flowed in after patterning. Finally, we program domain-specific hydrogel swelling in fabricated DNA-functionalized hydrogels by adapting recipes for DNA-crosslinked hydrogels from *Shi et al.* [18] that can undergo addressable swelling based on the DNA hairpins added to the solution surrounding the hydrogel. In summary, we demonstrate a hydrogel fabrication platform that can fabricate multi-domain DNA-functionalized hydrogel architectures capable of addressable hybridization and DNA-induced swelling.

## Materials and methods

A detailed protocol for MAPDH is published on protocols.io, https://dx.doi.org/10.17504/protocols.io.j8nlkw2ywl5r/v1 and is in printable form S1 File.

### MAPDH code

The Python script for Micro-Manager, available at https://github.com/MishaRubanov/MAPDH/, operates MAPDH. Micro-Manager controls a digital micromirror device (DMD), a UV LED source for photopatterning, and an XY stage for precise positioning during patterning. It also manages electronic valves for fluid flow control. The Python code is utilized to specify the sequence and duration of ink flow, as well as to define the UV exposure time, intensity, shape, size, and position of the patterned hydrogels. The microcontroller board, though initially mentioned, primarily serves to control the flow of solutions through the device, while the camera and the fluorescent LED, previously listed, are not directly relevant to the patterning process. In the results section, it is emphasized that Python script interfaces with essential MAPDH hardware, including the DMD, UV LED, microscope XY stage, and a custom-built pneumatic flow controller.

### DNA sequences and preparation

The DNA sequences used are listed in Section 34 in S1 File. All oligonucleotides were purchased from Integrated DNA Technologies, Inc. (IDT). All fluorophore-modified DNA strands were synthesized and purified under RNase-free conditions by high-performance liquid chromatography (HPLC) by IDT. All other DNA strands were prepared by standard desalting. The DNA crosslink strands were modified with an acrydite moiety for co-polymerization with the hydrogel.

### Hydrogel ink solution preparation

All hydrogels were prepared in conical vials (Fisher; 3431ANK) with a final volume of 100–200 µL. The base formula for these hydrogels was 10 v/v% poly(ethylene glycol) diacrylate (PEGDA) with 575 (Sigma-Aldrich; 437441) or 10K (MilliporeSigma; 729094) molecular weight (MW), 1 w/v% Lithium phenyl-2,4,6-trimethylbenzoylphosphinate (LAP; Allevi), and a 1X Tris-acetate EDTA buffer solution (Sigma-Aldrich; 1.06174) with 12.5 mM Mg2+ (1X TAEM). The specific concentrations of the fluorophore-modified DNA strands and crosslinks in each of the ink solutions are described in the Sections 6, 10, 14, 16, 21, 23, 25, 29 in S1 File.

### Sacrificial layer removal

We used 1 M of sodium chloride (NaCl; Fisher Chemical; 7647-14-5) solution to dissolve the sacrificial layer by converting the poly(acrylic acid) (PAA) into its water-soluble form via Na

+ ion exchange (Section 21 in S1 File). After 1 hour, we replaced the waste receptacle with a 1.7 mL microcentrifuge tube (VMR; 87003–294) to collect the hydrogels during the next wash step. We then washed with 1 mL of 1X TAEM solution to convectively remove detached hydrogels and collect them in the microcentrifuge tube.

### Single-domain DNA-crosslinked hydrogel swelling

The protocol for single-domain hydrogel swelling is described in the Section 26 in S1 File. System 1 DNA swelling signals (S1_H1 and S1_H2) were first diluted to 100 μM and then heated to 95°C to completely denature the single-stranded DNA and prevent undesired secondary structures. Immediately after the heated strands were extracted from the heating source, they were placed in an ice bath in order for the strands to form a secondary hairpin structure. After formation of the hairpins, we pipetted the DNA swelling signal solutions into the wells of a 96-well plate where we pipetted the hydrogels we attempt to swell. Each well ends up with a hairpin final concentration of 20 μM and a buffer solution concentration of 1X TAEM.

### Multi-domain DNA-crosslinked hydrogel swelling

Protocol for multi-domain hydrogel swelling is described in the Section 30 in S1 File. For the red domain swelling stage, S1_H1 and S1_H2 DNA solutions were first heated and cooled in an ice bath to form the hairpin structure. S1_H1 and S1_H2 DNA solutions were then pipetted into the well that contains the 3-domain hydrogel for a final concentration of 20 μM of S1 DNA swelling signals. After 24 hours of time-lapse fluorescence imaging, we repeated the above process using system 2 hairpins (S2_H1 and S2_H2) to actuate the blue domain hydrogel.

## Results

Here we present a new method for automated patterning of DNA-functionalized hydrogels, which we term **M**ulti-domain **A**utomated **P**hotopatterning of **D**NA-functionalized **H**ydrogels (**MAPDH**). MAPDH is a photopatterning method which fabricates architected DNA-crosslinked hydrogels within a microfluidic chamber (Fig 1A) using an integrated Python script (Fig 1B and Sections 1, 2 in S1 File) [31]. Python script interfaces with all MAPDH hardware–it is used to control a digital micromirror device (DMD), a UV LED, a microscope XY stage, and a custom-built pneumatic flow controller. A Python script can specify which vial to flow ink from (FLOW()), where to pattern (MOVE(DOMAIN[])), as well as the shape to pattern with (PATTERN(DOMAIN[])) (Sections 1, 2 in S1 File). The patterning method is sequential–we first flow in a particular ink, move the chamber to a specified location, upload a mask, then expose the chamber to UV light, which polymerizes the ink to form a hydrogel domain or domains. Each ink is a solution that consists of a monomer, a photoinitiator, and specific acrydite-labeled DNA strands. The DNA strands can also include modifications, such as a fluorophore that allows for visualization. After hydrogels with specified shapes and locations are patterned using a particular ink, an automated wash step is applied which removes the ink. Another flow step can then be started to flow in a new ink from a separate vial (Fig 1C). This process can be repeated and can be used for up to 4 different inks in 4 different vials. A schematic of this process is described using the above syntax in Fig 1B. A schematic of how this syntax describes the series of ink flow, patterning, and washing steps during MAPDH is shown in Fig 1C.

   MAPDH uses an electronically controlled, pneumatic, low-dead volume microfluidic flow controller to direct automatic influx of inks and washing fluid into a microfluidic chamber and controls how regions of the chamber are exposed to light using a digital micromirror array (Fig 1D and Section 3 in S1 File) [32]. Pressurized air provides the driving force (Fig 1D$_1$)

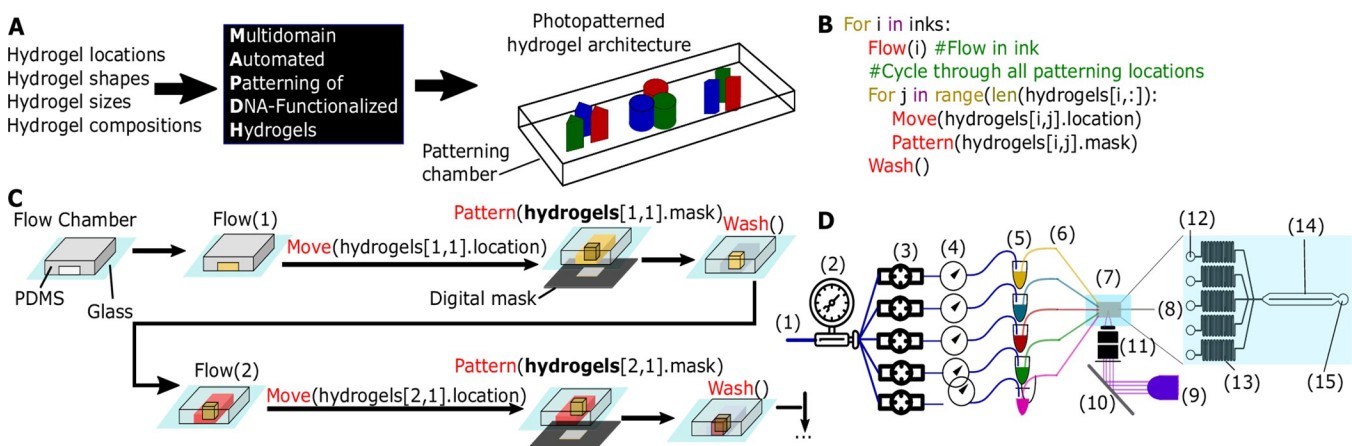

**Fig 1. Multidomain, Automated Photopatterning of DNA-functionalized Hydrogels (MAPDH). A)** Generalized workflow for fabricating DNA-functionalized hydrogels. The hydrogel location, size, shape, and composition are specified, after which an automated script is run, and the output is a chamber with photopatterned hydrogels of different sizes, shapes, and compositions. **B)** Pseudocode for MAPDH in Python. The algorithm takes as input the vials that will be flowed through the patterning chamber. The algorithm also takes as input the hydrogel location, as well as the mask (and thus the shape) of each hydrogel. The algorithm incorporates a wash step between each round of patterning. **C)** Workflow for MAPDH. After fabrication, inks are flowed and patterned sequentially, with an automated wash step between each patterning step. These domains can be patterned with interpenetrating borders, or as standalone hydrogels. **D)** MAPDH Setup. Pressurized air (1) is controlled by a pressure regulator (2) to 10 PSI. The air from the regulator is split into 5 different solenoid electronic valves (3), which are initially closed. When open, the air from one of the electronic valves is flown through a pressure gauge (4) and injected into an air-tight vial (5) above the ink. Another tube, located at the bottom of the conical vial, is filled with ink. When the electronic valve is turned on, the pressurized air pushes on the air inside the ink vial, which forces ink up the second tube (6) (Section 3 in S1 File). The second tube is fed directly into a microfluidic device (7), and after patterning the ink exits through an outlet into a waste receptacle (8). To photopattern hydrogels in place, we shine UV light (11) towards a digital micromirror device (10), where the light is reflected at specified locations where the mirrors are activated, which enables the shaping of light, and focused through an objective (9) onto the microfluidic patterning chamber. The microfluidic chamber has 5 inputs (12), followed by 5 microfluidic resistors (13), which prevent backflow (Section 4 in S1 File). The patterning chamber (14) consists of 2 rows for photopatterning, followed by an outlet (15).

directly fed into a pressure regulator (Fig 1D$_2$) that controls the pressure to 5 different electronically controlled pneumatic solenoid valves (Fig 1D$_3$). Pressure to each vial is precisely and modularly controlled using fine pressure gauges (Fig 1D$_4$). The fine pressure gauges are then connected directly to ink vials, and the pressure applied to the ink vials controls the corresponding flow rate of ink from that vial. The inks are housed in 1 mL air-tight conical vials with two drilled holes in the lid. Pressurized air from the pneumatic flow controller enters in an air input tube through one hole into the air above the ink inside the vial. The second hole houses an ink output tube where ink solution is output into the chamber when air pressure is applied to the vial through the air input tube (Fig 1D$_5$). By opening the solenoid valve, the ink is driven through the second tube and out of the vial. To minimize the dead volume for MAPDH of each ink the ink output tube starts at the very bottom of the conical vial, so that patterning occurs reliably when a vial contains as little as 100 μL of ink (Section 3 in S1 File). The ink output tube (Fig 1D$_6$) feeds directly into a microfluidic chamber (Fig 1D$_7$). The microfluidic chamber consists of annealed polydimethylsiloxane (PDMS) onto glass and has 5 input ports for 4 inks and 1 wash solution (Fig 1D$_{12}$ and Sections 4, 5 in S1 File). After photopatterning, wash solution flushes the cell, removing the ink. This waste exits the chamber through a waste tube (Fig 1D$_8$ and 1D$_{15}$). We expose the microfluidic chamber to UV light that is focused through an objective (Fig 1D$_9$) to achieve photopatterning resolution on the order of 10s of microns. The UV light (Fig 1D$_{11}$) is shaped using a digital mask uploaded to a Digital Micromirror Device (DMD) (Fig 1D$_{10}$) and the light is projected onto the chamber to precisely control hydrogel photopolymerization and shape. Microfluidic resistors prevent backflow from the microfluidic chamber into the ink output tubes (Fig 1D$_{13}$). The region of the chamber where patterning occurs consists of two chambers separated by a 100 μm thick PDMS wall (Fig 1D$_{14}$).

We initially characterized the shapes of hydrogels and the concentrations of DNA within PEGDA-co-DNA hydrogels fabricated using MAPDH. We set up a MAPDH protocol, which consists of multistep fabrication of DNA-functionalized hydrogels using an automated workflow and multiple vials (Section 6 in S1 File). We fabricated hydrogels containing fluorescent DNA and then measured the shapes of the hydrogels in fluorescence micrographs and the fluorescence intensities within the shapes. Initially, we fabricated 5 hydrogels with the same ink using MAPDH, with a target length of 50 µm. The resultant lengths for these 5 hydrogels are 51.5 µm ± 0.6 µm (mean ± std dev), which was within our goal for accurately fabricating hydrogels within a resolution of 10s of microns (Section 7 in S1 File).

We next sought to measure the variation in hydrogel shape and DNA concentration between patterning rounds, where a patterning round consists of 1) ink flow, 2) printing, and 3) washing. The printing step (#2) consists of one or more cycles of a) moving the chamber to a location, b) flowing the ink used in #1 for a few seconds, and c) photopatterning. The purpose of b) was that we found that flowing the ink needed for patterning for a few seconds just before photopatterning led to low variance in the patterned sizes of hydrogels (Section 6 in S1 File).

To measure the amount of variation in fabricated hydrogel size, we set up a MAPDH protocol with 12 patterning rounds, in which a column of 5 rectangular hydrogel posts were patterned in each round. The 12 rounds used VIALS = [1,2,3,4,1,2,3,4,1,2,3,4], respectively (Fig 2A–2G and Section 6 in S1 File). After each column of 5 hydrogels was patterned, the chamber was washed using the 1X TAEM buffer solution. There was no significant difference between the average lengths or fluorescence intensities of the hydrogels patterned in each round (Fig 2H and 2I and Section 7 in S1 File). Thus, fabricating hydrogels with different vials and/or inks using MAPDH can be done interchangeably without affecting hydrogel properties such as length or fluorescence.

We then explored the ability to fabricate hydrogels smaller than 50 µm in length: we were able to fabricate hydrogels using 20 and 10 µm length square masks with average lengths of

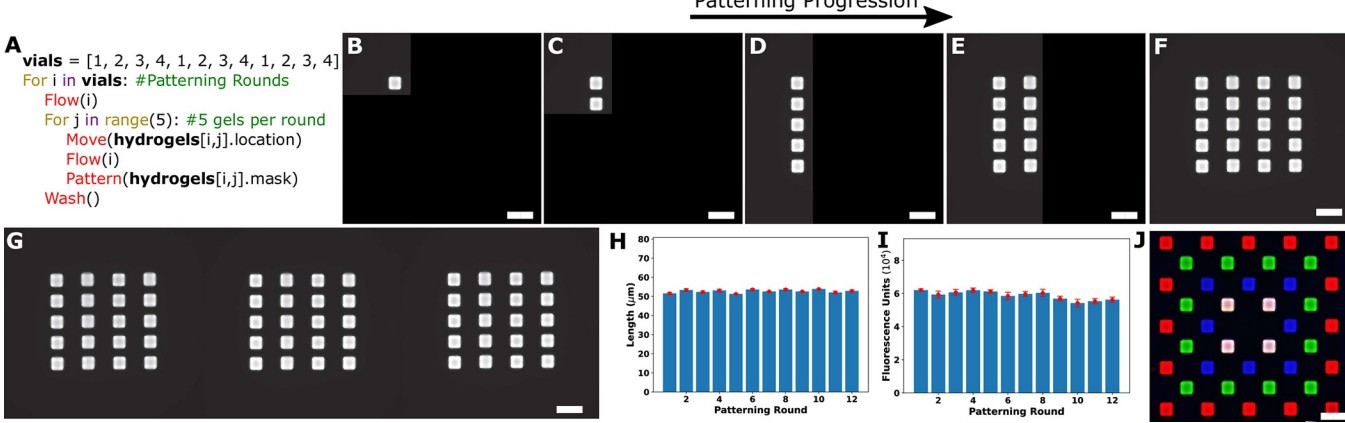

**Fig 2. MAPDH characterization. A)** Pseudocode describing automated workflow. 4 inks were patterned 3 times for a total of 12 rounds. Each round consists of patterning 5 rectangular hydrogels in a column before moving to the next column. B-F: Micrographs of the patterning progression. **B)** Patterning the first hydrogel during the first round. **C)** Patterning the second hydrogel during the first round. **D)** Completing the first patterning round. **E)** Completing the second patterning round. Round 1 used vial 1 and round 2 used vial 2, which both contained the same ink. **F)** Completing 4 patterning rounds. Vials 1, 2, 3 and 4, respectively were used during the 4 rounds. **G)** Completing all 12 patterning rounds using the program in A. **H)** Average length for each hydrogel within a round. Error bars are standard deviation in length per hydrogel within a round. **I)** Average Fluorescence for each hydrogel within a round. Error bars are standard deviation in fluorescence per hydrogel within a round. **J)** Combined micrograph of 4 different hydrogel domains using 4 ink compositions in separate vials. The inks each contained 100 nM of acrydite-modified, fluorophore-modified DNA, except for the white domain, which contained 100 nM of all 3 DNA strands. Ink compositions are shown in Sections 6–10 in S1 File. B-G were run using a MAPDH script in Section 6 in S1 File. Fig J was run using a separate MAPDH script in Section 10 in S1 File. Scale bars are 100µm.

27.1 ± 0.7 µm and 13.69 ± 0.7 µm (mean ± std dev), respectively, using a modified version of MAPDH (Sections 8, 9 in S1 File). Finally, we asked whether we could pattern multi-domain hydrogel systems containing different types of hydrogels: we added 4 different types of inks with 4 different acrydite- and fluorophore-modified DNA to 4 different vials; all the patterned hydrogels had specified fluorescence intensities in 3 different fluorescent channels (Fig 2J and Sections 10, 11 in S1 File).

To test the scalability of the MAPDH system, we next created a multi-domain hydrogel "image" consisting of individually patterned rectangular hydrogel "pixels". We chose a 15x15 pixel art image of a watermelon that contained four unique colors (Fig 3A). To develop a MAPDH protocol for this hydrogel architecture, we wrote an algorithm that constructed four 15x15 binary matrices specifying the location to pattern each color in the watermelon image, corresponding to four hydrogel domains (Fig 3B and Section 12 in S1 File). The locations extracted from the image were then converted into locations for patterning during MAPDH which were then automatically converted into a MAPDH protocol to construct the watermelon grid (Section 13 in S1 File). We observed no significant misalignment between the patterning of each individual hydrogel pixel–the hydrogels were precisely patterned at their specified locations, with no overlap between neighboring hydrogel pixels. Since we had three orthogonal wavelengths to image the various fluorophores in, the fourth ink (represented as pink in Fig 3A and 3B) was a combination of all three fluorophores. We false-colored the composite micrograph based on the colors in the original pixel-art image. In the final hydrogel architecture, dark green corresponded to 500nM of 5Acry_3Cy3_polyT10, light green corresponded to 500 nM of 5Acry_3ATTO488_polyT10, red corresponded to 500 nM of 5Acry_3-TYE665_polyT10, and pink corresponded to 500 nM of all three acrydite- and fluorophore-modified strands (Sections 14, 15 in S1 File).

Next, we asked whether hydrogels fabricated through a MAPDH protocol could, by controlling where DNA is conjugated within hydrogels in an architecture, control where hybridization reactions occur. This would make it possible to control where DNA circuits operate or

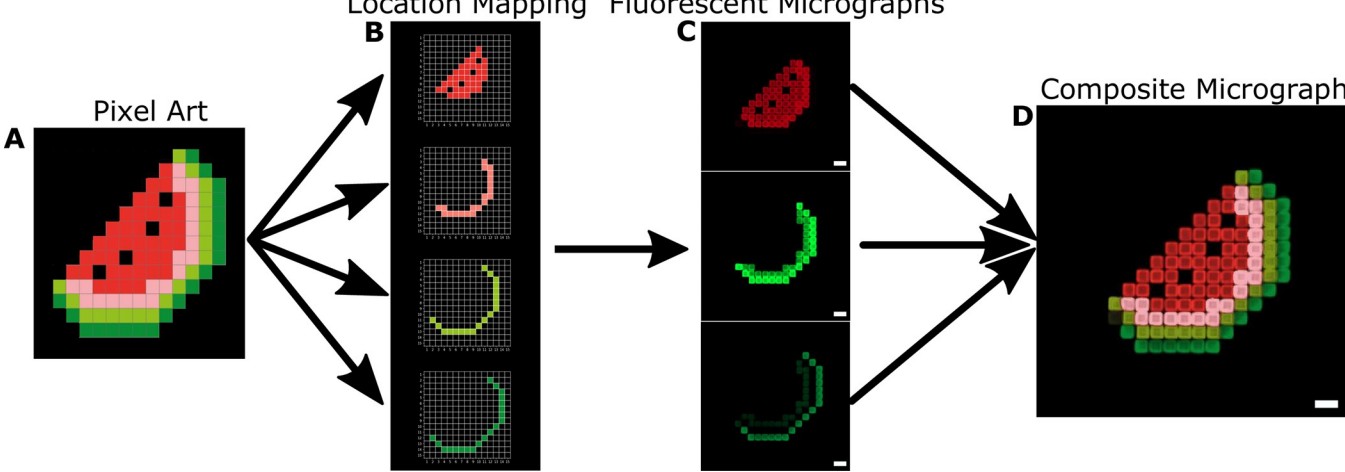

**Fig 3. Fabricating a pixel-art based 4-domain DNA-functionalized hydrogel. A)** We chose a 15x15 pixel art image consisting of four colors. **B)** Using an automated script, the four colors are distributed into four 15x15 binary matrices storing the locations for patterning of each domain. **C)** The four binary matrices are then fed into a MAPDH protocol designed to take as input a set of locations for each ink to pattern. In this case, the fourth ink contained a combination of all three fluorophores, resulting in hydrogels patterned in that domain visible in all three imaging channels. A colormap was applied to each fluorescent micrograph to match the color on the original pixel art image. **D)** The fluorescent micrographs are then combined for a composite micrograph shown in D, with the pink domain (the fourth ink patterned) colored based on fluorescence intensities from all three fluorescent micrographs. Scale bars are 100µm.

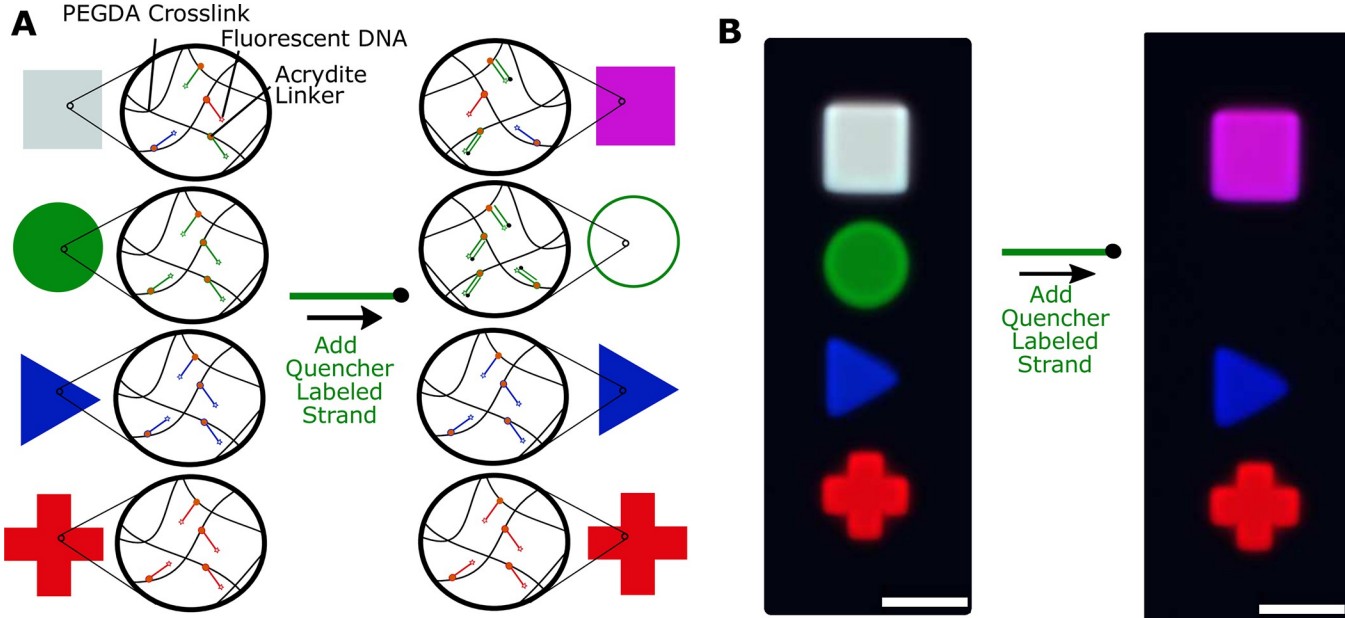

**Fig 4. Addressable hybridization reactions in MAPDH-fabricated hydrogel architectures. A)** Schematic illustrations of each domain in the hydrogel architecture showing the acrydite- and fluorophore-modified DNA each domain contains. The arrangement of the domains in the illustration is also their designed arrangement in the fabricated hydrogel architecture. The ink for the circular hydrogel post contained 500nM of 5Acry_3Cy3_R1 (labeled as a green ink), the ink for the triangular hydrogel post contained 500 nM of 5Acry_3ATTO488_polyT10 (labeled as a blue ink), the ink for the plus-shaped hydrogel post contained 500 nM of 5Acry_3TYE665_polyT10 (labeled as a red ink), and the ink for the square hydrogel post contained all 3 of the listed DNA strands, each at 500 nM. The fabricated circular and square posts thus contained DNA that could hybridize to quencher-modified DNA strand (5Q_R1'), the complement of 5Acry_3Cy3_R1, when solution containing 5Q_R1' is added to the chamber (sequences in Section 34 in S1 File). **B)** Composite multicolor fluorescence micrographs before and after addition of solution containing 500 nM 5Q_R1'. The square post appears white before hybridization because it has high green, blue and red fluorescence intensity, but magenta after adding the solution containing 5Q_R1', because it has high blue and red fluorescence intensity but less green fluorescence intensity. Scale bars are 100 μm.

where other DNA-mediated downstream processes take place [4, 12, 14, 33]. We fabricated a hydrogel architecture with 4 hydrogels, each a different domain. One of the hydrogels in the architecture, a circle, contained a fluorescently labeled DNA molecule 5Acry_3Cy3_R1. A second hydrogel, a square, contained 5Acry_3Cy3_R1 and two other DNA species labeled with Atto488 and TYE665. The two other hydrogels, a triangle and a plus sign, had other DNA species labeled with Atto488 and TYE665 (Fig 4A). The fluorescence produced by 5Acry_3-Cy3_R1 could be quenched through hybridization by a complementary sequence bearing a fluorescence quencher, 5Q_R1' (Section 16 in S1 File).

After this hydrogel architecture was fabricated, a solution containing 5Q_R1' was flowed into the microfluidic chamber. We then observed that the hydrogel that contained only 5Acry_3Cy3_R1 (the green circle) decreased in fluorescence intensity and the fluorescence intensity of the hydrogel containing all three fluorophore-modified DNA strands decreased in the channel corresponding to the 5Acry_3Cy3_R1 label (Cy3), but not in the other two observed fluorescence channels. The fluorescence intensities of the other two hydrogels did not change after 5Q_R1' was added to the microfluidic chamber (Fig 4B and Section 17 in S1 File). Thus, DNA in solution can hybridize with DNA conjugated to MAPDH-fabricated hydrogels, enabling addressable control of DNA hybridization within hydrogel architectures.

We next sought to develop a method of fabricating free architected hydrogels by fabricating hydrogels in a microfluidic device, lifting them off the surface, removing them from the device and collecting them. We call this method MAPDH-LC, or Multi-Domain Automated Photo-patterning of DNA-Functionalized Hydrogels with Lift-off and Collection. We began by

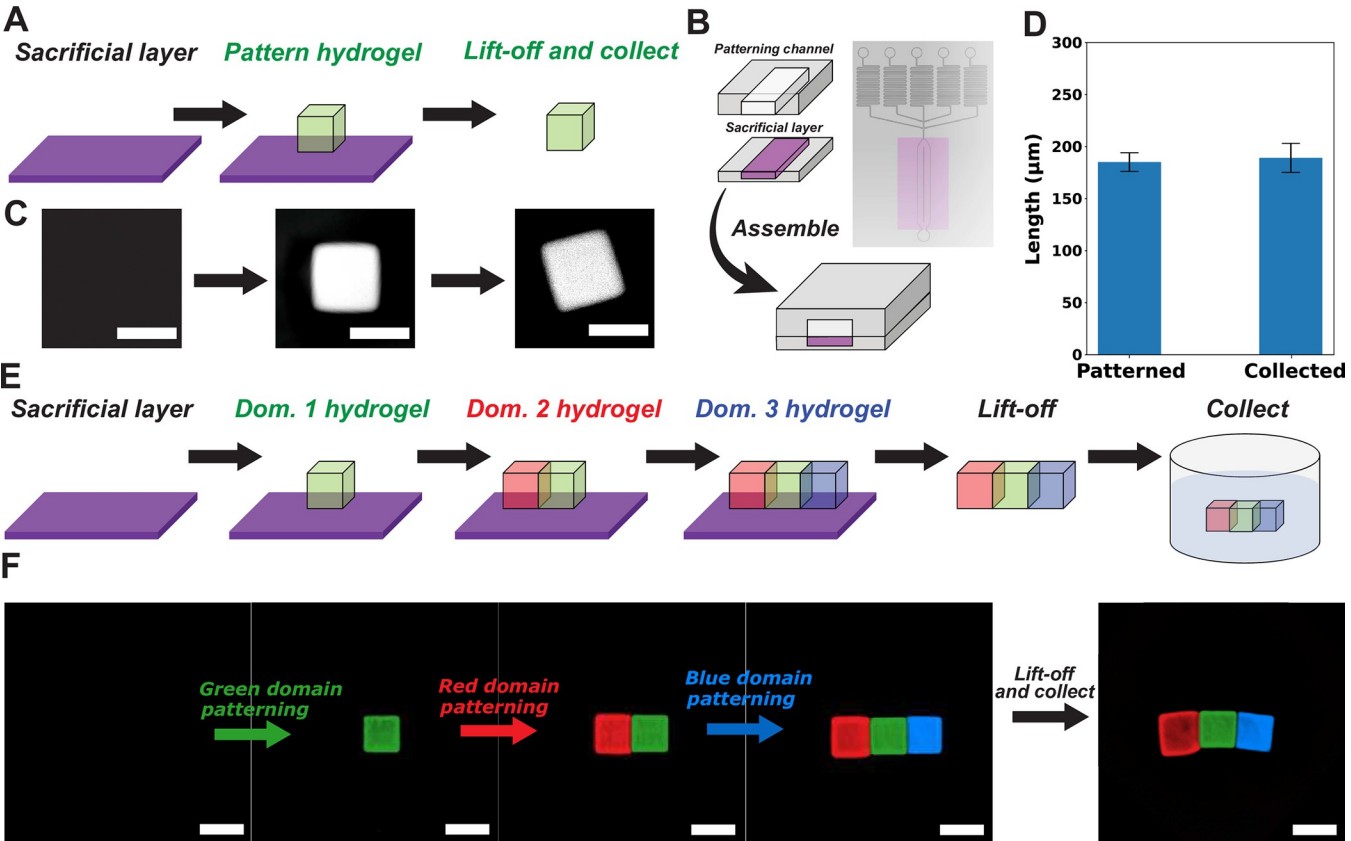

**Fig 5. Lift-off and collection of single-/multi-domain hydrogels using MAPDH-LC. A)** Schematic illustration of single-domain hydrogel patterning and lift-off. **B)** Schematic illustration of the multi-domain patterning chamber assembled with the sacrificial layer (Section 20 in S1 File). **C)** Representative images of single domain ink flow, patterning, and washing. Scale bars are 200 μm (Section 21 in S1 File) **D)** Length measurements of single-domain hydrogels after patterning and after collection (N = 7). Error bars indicate standard deviation in hydrogel length for seven independent measurements (Section 22 in S1 File). **E)** Schematic illustration of multi-domain hydrogel patterning and lift-off for collection. **F)** Profiles of multi-domain hydrogel patterning and lift-off for collection (Sections 23, 24 in S1 File). Scale bars are 200 μm.

developing MAPDH-LC for fabricating single-domain hydrogels and then detaching them from the device surface.

This method uses a sacrificial layer that, after patterning, is controllably dissolved to lift-off and remove hydrogels from a modified microfluidic chamber, termed the lift-off flow chamber (Fig 5A). Prepolymer solution for the sacrificial layer is poured into a PDMS 'boat' for thermal curing (Sections 18, 19 in S1 File). After thermal curing, the PDMS boat with the sacrificial layer is sandwiched with a second PDMS structure containing the patterning chamber to form a lift-off flow chamber (Fig 5B and Section 20 in S1 File). We chose PAA for the sacrificial layer due to PAA's ability to selectively dissolve in NaCl [34].

To test this method, we first fabricated 25 simple rectangular PEG hydrogels containing 5Acry_3Cy3_polyT10, so that the resulting hydrogels would be fluorescent (Fig 5C). After patterning, a 1 M NaCl solution automatically flowed into the chamber to detach the hydrogels from the sacrificial layer within the chamber. To extract the detached hydrogels, we flowed the same buffer solution used in the wash steps, 1X TAEM, to remove the hydrogels from the device *via* convection (Section 21 in S1 File).

After the free hydrogels were extracted from the device, we developed a protocol for collecting the hydrogels that involved concentrating the hydrogels in a small volume of buffer before

transfer into a 96-well plate. The hydrogels that had been removed from the device were washed into a microcentrifuge tube with 1 mL of 1X TAEM buffer solution. The microcentrifuge tube containing the hydrogels was then centrifuged for a few seconds so that the hydrogels were concentrated at the bottom of the tube. After centrifugation, 20 µL of the bottom of the solution was pipetted into a 96-well plate for observation and further analysis (Section 21 in S1 File).

After collection, we observed 13 free hydrogels within the 96 well plate. Of those 13 hydrogels, we measured the lengths of one side for the 7 hydrogels that lay flat on the surface of the 96-well plate (Section 22 in S1 File). We compared the lengths of the 7 hydrogels measured after collection to the side lengths of 7 randomly selected hydrogels within the chamber just after patterning (*i.e.*, before liftoff) (Fig 5D). The side length (mean ± std dev) of the hydrogels after lift-off and collection was 189 ± 14 µm, and the mean side length of hydrogels just after patterning was 195 ± 9 µm. These results suggest that this process could be used to fabricate free hydrogels of a specified size.

We next asked whether we could extend MAPDH-LC to fabricate free multi-domain hydrogels. We considered a target three-domain hydrogel architecture composed of three adjacent 200x200 micron domains (Fig 5E and Section 23 in S1 File). We organized the domains so that adjacent patterned areas each overlapped by 10 microns.

We first asked whether we could successfully lift three-domain hydrogels off the device surface after performing multiple rounds of patterning using MAPDH. We attempted to fabricate 25 three-domain hydrogels in three patterning rounds where we patterned each of the three types of domains. We observed that 12 of the 25 three-domain hydrogels were washed away during one of the three patterning rounds. The 13 hydrogels that remained after the third patterning round were removed from the device surface during the 1 M NaCl wash step (Section 23 in S1 File).

To recover these three-domain hydrogels (similar to recovery of single domain hydrogels) using this process, we patterned the domains of the hydrogels with small overlapping regions. We wanted to see whether a 10-micron overlap was sufficient to keep the hydrogel domains together during the lift-off and collection process. Of the 13 three-domain hydrogels that remained adhered to the PAA surface before lift-off, we successfully collected two into a 96-well plate (Section 24 in S1 File). We observed that for both hydrogels collected, the three domains remained attached to each other after collection, suggesting that the interfaces created during patterning were mechanically strong enough to prevent breaking during lift-off and collection (Fig 5F).

DNA-crosslinked PEG hydrogels can swell in response to specific DNA sequences [14]. We asked if we could use MAPDH-LC to fabricate DNA crosslinked hydrogels and whether these hydrogels would swell in response to DNA sequences. Thus, in order to fabricate DNA-crosslinked hydrogels using MAPDH-LC, we adapted an ink formulation used for photopatterning DNA-crosslinked PEGDA-10K hydrogels from Shi *et al.* [18] (Fig 6A). We changed the photoinitiator and exposure time, while keeping the monomer (PEGDA-10k), DNA sequences, and DNA concentrations identical. We replaced the photoinitiator originally used in that work with Lithium phenyl-2,4,6-trimethylbenzoylphosphinate (LAP) and the exposure time was reduced to 1 second (Section 25 in S1 File).

Using this ink, we were able to fabricate well-formed hydrogels, lift them off, and collect them. Of the 30 hydrogels that were photopatterned, 18 hydrogels were collected. Of these 18 hydrogels, 11 lay flat on the surface (without bending or wrinkling) in the 96-well plate where the hydrogels were collected, meaning that if they swelled, the extent of swelling could be measured easily by measuring the changes in their lengths (S19 Fig). We next swelled the hydrogels and used the 11 that laid flat to measure swelling extent over time (Section 26 in S1 File).

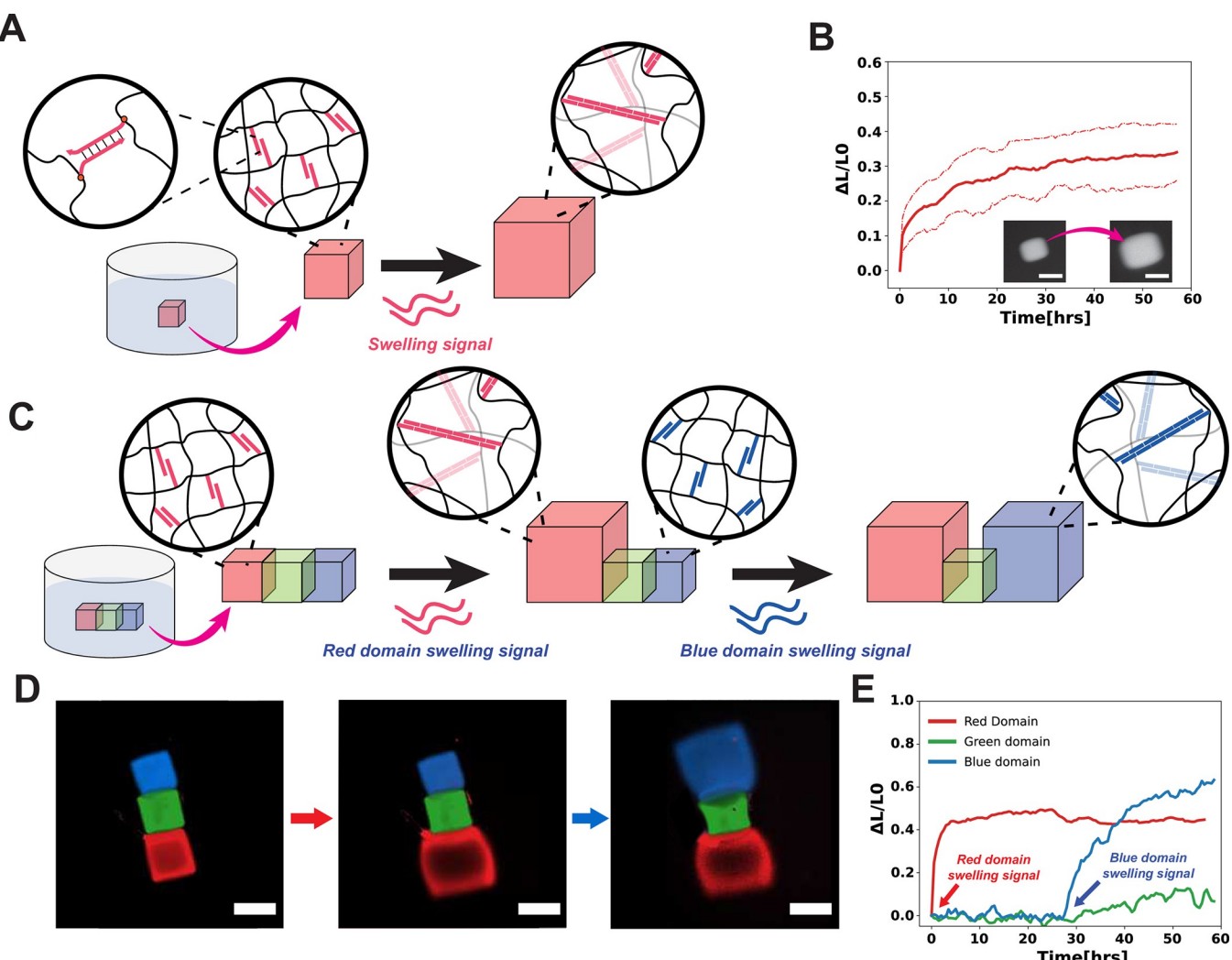

**Fig 6. Swelling of collected single-/multi-domain hydrogels using MAPDH-LC. A)** Schematic illustration of swelling single-domain hydrogels. **B)** Swelling profile of single-domain hydrogels (N = 7). Inset shows micrographs of the hydrogels before swelling signal is added, then 60 hours after the swelling signal has been added. Dotted lines indicate the minimum/maximum values of seven independent sample measurements. Scale bars are 50 μm (Sections 25–28 in S1 File). **C)** Schematic illustration of swelling multi-domain hydrogels. **D)** Micrograph images of the multi-domain hydrogels after the red and blue domains undergo swelling via addition of red and blue swelling signals, respectively. Scale bars are 200 μm. **E)** Swelling profiles of each hydrogel domain after distinctive swelling signals have been added within 60 hours (Sections 29–31 in S1 File).

To swell the hydrogels, we added two DNA swelling signals (System 1 hairpins, S1_H1 and S1_H2) at 20 μM each to the solution surrounding the free hydrogels, initiating the DNA hybridization chain reaction that drives hydrogel swelling [14]. We measured the amount of swelling of 11 single-domain hydrogels over 60 hours in terms of the lengths of the hydrogels at a given time divided by their original lengths, ($\Delta L/L_0$) (Fig 6B). The $\Delta L/L_0$ after 60 hours was 0.51 ± 0.010 (mean ± std dev) (Fig 6B), which is slightly larger than the $\Delta L/L_0$ of approximately 0.4 after 60 hours reported in Shi *et al.* [18] (Sections 27, 28 in S1 File). Thus, MAPDH-LC can be used to fabricate and collect single-domain DNA-crosslinked hydrogels capable of swelling in response to specific DNA sequences.

We next attempted to use MAPDH-LC to fabricate and collect 3-domain DNA-crosslinked hydrogels where different domains within this structure swell in response to different DNA sequences (Fig 6C). Two of the domains had two different DNA-crosslinks within them (red

and blue in Fig 6C), while the third domain did not contain any DNA crosslinks. Thus, the red domain should swell in response to one sequence (termed the red swelling signal), the blue domain to another sequence (termed the blue swelling signal), and the green domain should not swell in response to any DNA stimuli. We used MAPDH-LC (Fig 5E) to pattern 50 three-domain hydrogels. We were able to successfully collect just one free 3-domain hydrogel (Section 29 in S1 File).

To test whether the patterned domains were selectively responsive, we actuated the red and blue domains sequentially by adding in red and then blue swelling signals. We observed swelling of the red domain for the first 10 hours after addition of the red DNA swelling signal (system 1 hairpins) (Fig 6D). After about 24 hours, we added the blue DNA swelling signal (system 2 hairpins) to initiate swelling of the blue-domain hydrogel (Fig 6D and Section 30 in S1 File). We observed that the swelling of the red and blue domains was constricted near the edge shared with the green domain. To quantify the amount of swelling, we chose the edge opposite the green domain to measure $\Delta L/L_0$. After 60 hours, this $\Delta L/L_0$ was 0.4 for the red domain and 0.6 for the blue domain (Fig 6E and Section 31 in S1 File), consistent with the swelling observed for single-domain hydrogels. The green domain did not appear to swell, but it bowed outward over time near its edges contacting the swelling domains (Fig 6D), likely due to the stress applied by the swelling of the blue and red hydrogel domains. We also measured the differences in extent of swelling between the edges of the red domain closest and farthest from the green domain, which was 0.217. This was approximately 50% of the final $\Delta L/L_0$ observed, suggesting some control over swelling within a hydrogel domain by constraining regions of that domain with non-swellable domains. These results indicate how control and restriction of swelling is possible by patterning DNA-crosslinked domains that swell in response to different signals using MAPDH-LC. Because MAPDH-LC can be used to pattern complex multi-domain architectures, MAPDH-LC could be used to readily fabricate hydrogels that programmatically morph into different 2D- or 3D- shapes.

## Discussion

Here we demonstrated an integrated method for multi-domain DNA-functionalized hydrogel fabrication, MAPDH. We could reproducibly pattern hydrogel domains in multiple rounds and fabricate hydrogels with shapes and compositions with sizes ranging from 10s to 100s of microns. We showed the ability to fabricate 4-domain DNA-functionalized hydrogels to recreate a pixel art image with precise alignment between hydrogel pixels. DNA oligos can diffuse into patterned hydrogels, where they can engage in local hybridization reactions with DNA functionalized to the patterned hydrogels. Finally, we developed a method for selectively removing the hydrogels from the substrates after patterning using a dissolvable sacrificial layer. We demonstrated the ability to pattern and remove a three-domain interconnected DNA-functionalized hydrogel architecture from the patterning chamber and selectively swell hydrogel domains using a hybridization chain reaction in a separate container.

Conventionally, hydrogels with various functionalizations, including DNA, have been patterned using a photomask [12] or a replication mold [35], but these fabrication methods require masks with pre-defined shapes and sizes. Recent studies have integrated a photopatterning apparatus (e.g. a digital micromirror device or DMD) with an optical microscope setting for precise control of shape, position, and size of functionalized hydrogels [17, 36–38]. However, these methods involve manual ink injections, UV exposure, and manually positioning the substrate in the desired location for patterning which entails accurately placing the substrate, adding an additional layer of time and labor to the process. The manual performance for each of these steps multiple times during multi-domain patterning would be slow and

labor-intensive. Furthermore, existing methods require milliliter volumes of inks (*i.e.*, pre-gel solution) to allow for washing and filling a device, even though only a few microliters of solution end up in a final product. This requirement can make these methods costly, especially when these inks involve high concentrations of modified DNA strands. MAPDH streamlines the hydrogel patterning process with automated control of ink flow, patterning, and washing, and requires much lower volumes of inks (100 μL). While the DMD itself doesn't consume ink, the ink volume may be relevant to the overall fabrication process. MAPDH's automated control minimizes wastage, allowing for precise ink application and reducing costs associated with ink usage.

MAPDH is currently capable of printing 4 unique DNA-functionalized hydrogel domains, with a 5[th] vial required for washing the patterning chamber between patterning steps. Due to the modular nature of the flow controller, we envision future versions of MAPDH having much greater numbers of inks. To expand the number of inks, the pneumatic flow controller can be expanded, and the microfluidic chamber can be redesigned to accommodate a greater number of inlets. A future version of MAPDH could have 10s or even 100s of inks patterned automatically by incorporating a robotic fluid delivery system [39]. Additionally, we currently have the ability to control 2 dimensions when fabricating these hydrogels. Future directions also include expanding into the third stage, either by including a z-stage or through grayscale patterning [40].

Despite successful demonstrations of a method that can be used to photopattern multi-domain DNA-functionalized hydrogels, with lift-off via dissolution of the sacrificial layer and collection in an unconstrained environment, limitations remain in the lift-off and collection processes. One noticeable problem during collection is the misalignment of different hydrogel domains when patterning multiple domains (S20 Fig). This may have been caused by the weak anchoring of the hydrogel to the sacrificial layer, which causes movement of the hydrogel when ink or wash solutions are flown into the device. Sometimes the adhesion of the hydrogels to the sacrificial layer is so poor that the hydrogels patterned in one round are completely removed prematurely during subsequent patterning and wash steps, rather than during the lift-off stage. Our results indicate a relatively low yield of approximately 2% collection of 3-domain hydrogels in a single, multi-step patterning process (Fig 5E and 5F). Thus, future developments of MAPDH-LC could be focused on building a robust sacrificial layer that can anchor photopatterned hydrogels with strong adhesion so that minimal misalignments occur during flow steps. Additionally, inhomogeneous photopolymerization of hydrogels may result in non-uniform expansion in response to an added swelling signal [41] (Fig 6D). MAPDH could be optimized to achieve uniform photopatterning of hydrogels by minimizing UV exposure time and intensity.

MAPDH could be used to help fabricate multi-domain DNA-functionalized hydrogels capable of complex shape change [42]. Due to the precise control of size, location, and shape of the hydrogels, we envision that MAPDH could be used for local sequestration and release of molecules within hydrogels, either via timed release or designed reactions [6], to direct cell growth [43] or for local self-assembly [44]. Harnessing the biocompatibility and permeability of hydrogels, MAPDH may also be promising in various biomedical applications, such as designing controlled drug delivery systems [18], scaffolds [45], or organ-on-a-chip devices for tissue engineering [46].

## Supporting information

**S1 File. Step-by-step protocol downloaded from protocols.io.**
(PDF)

**S1 Fig. Vial setup and terminology.** Pressurized air is fed into the top of the air-tight vial through the air tube, which forces ink through the ink/wash tube out of the vial and into the microfluidic flow chamber.
(TIF)

**S2 Fig. Design of the multi-domain hydrogel patterning flow chamber.** This microfluidic flow chamber was designed using AutoCAD. The design was used to generate a mask (5) produced by Fineline Imaging.
(TIF)

**S3 Fig. Brightfield micrographs of hydrogels patterned at different z-planes.** The labels below each hydrogel are the percentages of a full rotation of the microscope, relative to focusing the camera on the glass surface-PDMS interface that the objective was set to before patterning. Since the DMD and camera are in different locations within the microscope and thus have different planes where they are in focus, we chose the focal plane for patterning by first focusing on the z-plane of the camera, and then adjusting the z-plane to match the ideal plane for patterning. **A)** 10x patterning and imaging. We observed that the sharpest edges were made when patterning at a plane 50% of a full turn below the plane where the camera is focused on the glass: PDMS interface of the microfluidic flow chamber. **B)** Patterning and imaging using the 20x objective. We observed that the sharpest edges were found when patterning at a plane 30% of a full turn below the plane where the camera is focused on the glass: PDMS interface of the microfluidic flow chamber. Scale bars are 50 μm.
(TIF)

**S4 Fig. Brightfield micrographs of hydrogels patterned as part of the single-domain hydrogel patterning experiment.** The first micrograph shows the hydrogels produced within a single round of patterning (Set 1). The last three micrographs show the resulting hydrogels from all 12 rounds of patterning, in order from left to right. Scale bars are 50 μm.
(TIF)

**S5 Fig. Micrograph overlay using ImageJ. A)** Bright field image **B)** Fluorescence image (Cy3) **C)** Overlay with brightfield grayscale and Cy3 in red. Scale bars are 100 μm.
(TIF)

**S6 Fig. Hydrogel measurements. A)** Brightfield micrograph for a single hydrogel. The outline was chosen manually using ImageJ's rectangle selection tool. The width and height of this rectangle corresponded to the width and height of the hydrogel. **B)** Overlayed brightfield and fluorescent Cy3 micrographs for a single hydrogel. The rectangle, selected from the brightfield micrograph in S6A Fig, is used to calculate the average fluorescence intensity for the hydrogel. Scale bars are 10 μm.
(TIF)

**S7 Fig. Small hydrogel patterning. A)** Hydrogels patterned automatically using 20 μm square masks and Inks 1–4 following the Section 9 in S1 File protocol. **B)** Hydrogels patterned automatically using 10 μm square masks and Inks 1–4 following the Section 9 in S1 File protocol. Scale bars are 20 μm.
(TIF)

**S8 Fig. Cy3 (red channel), ATTO488 (green channel), and TYE665 (blue channel) micrographs overlaid showing patterning results from Fig 2J experiment using the protocol in Section 10 in S1 File. A)** Architecture 1, **B)** Architecture 2, **C)** Architecture 3, **D)** Architecture

4. Scale bars are 100 μm.
(TIF)

**S9 Fig. Binning a pixel art image into a 15x15 grid.** The left image is a 355x355 pixel image, the right is a binned 15x15 version of the same image.
(TIF)

**S10 Fig. Color histogram for the watermelon pixel art image.** The length of each color bar represents the number of pixels with that corresponding color. The colors were clustered automatically using the k-means clustering method.
(TIF)

**S11 Fig. Binary masks for all four domains to be photopatterned using MAPDH.** These masks, termed location maps, serve as inputs for the MAPDH function matrix_patterner in Section 13 in S1 File.
(TIF)

**S12 Fig. Coloring grayscale micrographs based on original pixel-art colors.**
(TIF)

**S13 Fig. Multi-fluorophore domain.** Enhanced and colorized.
(TIF)

**S14 Fig. Hydrogels after patterning using the protocol in Section16 in S1 File, but before adding the strand 5Q_R1'.** The TYE665 (red channel), Cy3 (green channel), and ATTO488 (blue channel) micrographs are overlaid as described in Section 17 in S1 File. **A)** Set 1 **B)** Set 2 **C)** Set 3 **D)** Set 4 as described in Section 16 in S1 File. Scale bars are 100 μm.
(TIF)

**S15 Fig. Hydrogels after patterning and adding the strand 5Q_R1' following the protocol described in Section 17 in S1 File.** TYE665 (red channel), Cy3 (green channel), and ATTO488 (blue channel) micrographs overlaid as described in Section 17 in S1 File. **A)** Set 1 **B)** Set 2 **C)** Set 3 **D)** Set 4 as described in Section 16 in S1 File. Scale bars are 100 μm.
(TIF)

**S16 Fig. Normalized mean fluorescence counts for hydrogels in Fig 4 and S14B and S15B Figs. A)** Normalized fluorescence for the plus, triangle, and circle hydrogels in the respective channels that contained fluorescence. **B)** Normalized fluorescence for all three fluorophore channels for the square hydrogel.
(TIF)

**S17 Fig. CAD mask design for the PDMS boat.**
(TIF)

**S18 Fig. Demonstration of hydrogel production. A)** Photopatterning and washing. **B)** Lift-off and washing. **C)** Collection in a well of a 96-well plate. Top and bottom micrographs are two different locations from the same round of single-domain hydrogel patterning. Scale bars are 200 μm.
(TIF)

**S19 Fig. Representative micrograph of single-domain hydrogels fabricated using MAPDH-LC collected in a well of a 96-well plate.** Hydrogels on their sides and/or not flat on the glass surface are enclosed by dashed, pink circles. Only hydrogels that lay flat on the

bottom surface of the plate were further analyzed (top hydrogel).
(TIF)

**S20 Fig. Example multi-domain hydrogels fabricated using MAPDH-LC.** Out of the 13 aligned hydrogels 2 were successfully collected. **A)** Hydrogels in the microfluidic device. **B)** Hydrogels collected and placed in a well of a 96-well plate. **C)** Unaligned multi-domain hydrogels due to poor adhesion between the hydrogel and the sacrificial layer. Scale bars are 200 μm.
(TIF)

**S21 Fig.** Demonstration of measuring hydrogel lengths and comparison of (left) manual and (right) automated measurements. Herein, to automatically measure the hydrogel length we binarized the micrograph by a threshold calculated via Otsu's method. (Reference to MAPDH script).
(TIF)

**S22 Fig. Analysis of single-domain hydrogel swelling. A)** Original time-lapse micrograph of single-domain MAPDH hydrogel swelling. **B)** Automated micrograph analysis of single-domain MAPDH hydrogel swelling and measurement of length changes. **C)** Manual measurements of single-domain MAPDH hydrogel swelling and length change.
(TIF)

**S23 Fig. Analysis of multi-domain hydrogel swelling.** Micrographs of multi-domain hydrogel swelling after addition of red and blue swelling signals.
(TIF)

## Acknowledgments

The authors wish to thank Ruohong Shi and Kuan-Lin Chen for helpful advice and discussions.

## Author Contributions

**Conceptualization:** Moshe Rubanov, Rebecca Schulman.

**Data curation:** Moshe Rubanov, Joshua Cole, Heon-Joon Lee.

**Formal analysis:** Moshe Rubanov, Rebecca Schulman.

**Funding acquisition:** Rebecca Schulman.

**Investigation:** Moshe Rubanov, Joshua Cole, Heon-Joon Lee.

**Methodology:** Moshe Rubanov, Joshua Cole, Heon-Joon Lee, Zachary Chen, Elia Gonzalez.

**Project administration:** Rebecca Schulman.

**Resources:** Rebecca Schulman.

**Software:** Moshe Rubanov.

**Supervision:** Rebecca Schulman.

**Visualization:** Moshe Rubanov.

**Writing – original draft:** Moshe Rubanov, Joshua Cole, Heon-Joon Lee, Rebecca Schulman.

**Writing – review & editing:** Moshe Rubanov, Joshua Cole, Heon-Joon Lee, Leandro G. Soto Cordova, Rebecca Schulman.

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
