## [Decision Letter · Decision Letter 0]

12 Sep 2023

PONE-D-23-06068Multi-domain Automated Patterning of DNA-Functionalized HydrogelsPLOS ONE

Dear Dr. Schulman,

Thank you for submitting your manuscript to PLOS ONE. After careful consideration, we feel that it has merit but does not fully meet PLOS ONE’s publication criteria as it currently stands. Therefore, we invite you to submit a revised version of the manuscript that addresses the points raised during the review process.

We look forward to receiving your revised manuscript.

Kind regards,

Nileshkumar Dubey

Academic Editor

PLOS ONE

Journal Requirements:

3. Please remove your figures from within your manuscript file, leaving only the individual TIFF/EPS image files, uploaded separately. These will be automatically included in the reviewers’ PDF

Reviewers' comments:

Reviewer's Responses to Questions

**Comments to the Author**

1. Does the manuscript report a protocol which is of utility to the research community and adds value to the published literature?

Reviewer #1: Yes

Reviewer #2: Yes

2. Has the protocol been described in sufficient detail?

To answer this question, please click the link to protocols.io in the Materials and Methods section of the manuscript (if a link has been provided) or consult the step-by-step protocol in the Supporting Information files.

The step-by-step protocol should contain sufficient detail for another researcher to be able to reproduce all experiments and analyses.

Reviewer #1: Yes

Reviewer #2: Partly

3. Does the protocol describe a validated method?

Reviewer #1: Yes

Reviewer #2: Yes

4. If the manuscript contains new data, have the authors made this data fully available?

Reviewer #1: Yes

Reviewer #2: Yes

**5. Is the article presented in an intelligible fashion and written in standard English?**

Reviewer #1: Yes

Reviewer #2: Yes

6. Review Comments to the Author

Reviewer #1: The manuscript outline, together with SI and online sources, a detailed protocol for fabrication of hydrogel structures on up to four different precursor solutions in a photoinduced patterning/printing process. The selected proof of concept exploit DNA functionalized hydrogels that display specific stimuli responsive behavior that in the resulting matrices is demonstrated to be selective for a subdomain of a composite soft material structure, or selective. Overall, the presentation is detailed in the way it is outlining the whole value chain of the process from the hardware exploited, the control of the hardware, details of solutions etc. It is thus considered to be of substantial interest to the soft materials community in general and DNA-hydrogel experts in more specific. Nevertheless, some minor issues should be addressed before a final recommendation to publish can be given.

The first of this is the apparent inhomogeneity of the fluorescence intensity within domains of the hydrogels – e.g. in red domains in Fig 6D. What is the reason for such an inhomogeneity, and why does it appear more accentuated for the “red hydrogels” than for the other fluorophores?

Please check figure numbering – there appear to be two figures nr 4, and figure 6 referred to in the text is understood as figure 5 of the manuscript (this is assumed in the above statement).

There appear to some issue on presentation of references (e.g. Ref 4: Update year and add journal; Ref 9: There is an ACS Nano publication in 2020 with the same title – is that the one? Ref 10: missing year)

Reviewer #2: The purpose of the manuscript is to describe a new platform to fabricate DNA-functionalized hydrogels. The topic is relevant and up-to-date, including the use of computational models to design and control the reproducibility of the printed patterns. However, the manuscript should be better organized with an improved discussion of the presented data. Although the authors are providing the full protocol as supplemental materials, a brief description of the methods should be included in the main document to facilitate understanding for the reader. Also, since there is no discussion section, the results should be discussed and compared with the literature in the results section. The results section presented in the current version is purely descriptive with no evident or very limited comparisons with the literature, and, most importantly, it lacks explanations about the meaning of each finding. The limitations of the study and the further applications of the method could also be better addressed at the end of the results section.

Additional comments:

The authors target functionalization with DNA, but the platform provides many more options than that. Those possibilities should be better explored in both the introduction and the discussion. You are proposing a method with various applications in biomedical engineering and tissue engineering. So, it could be interesting to reinforce that in your rationale.

Define abbreviations upon their first appearance in the document.

Essential steps such as hydrogel preparation, DNA sequences, and functionalization should be included in the methods section. Although there is no need to describe the full code, it would be interesting to mention the language used for programming in the methods (Fig 1 and its description could be moved to the methods). Also, the swelling method and the sacrificial mold removal should be described in the main manuscript.

Even though the authors claimed the method is simple, the complexity of developing microfluidic systems, the alternated crosslinking in microscale, and the need for postprocessing to remove the sacrificial molds are not simple. Therefore, the idea of the “simple method” should be interpreted with care.

7. PLOS authors have the option to publish the peer review history of their article (what does this mean?). If published, this will include your full peer review and any attached files.

Reviewer #1: No

Reviewer #2: No

---

## [Author Response · Author response to Decision Letter 0]

27 Nov 2023

Reviewer #1: 

1) The manuscript outline, together with SI and online sources, a detailed protocol for fabrication of hydrogel structures on up to four different precursor solutions in a photoinduced patterning/printing process. The selected proof of concept exploit DNA functionalized hydrogels that display specific stimuli responsive behavior that in the resulting matrices is demonstrated to be selective for a subdomain of a composite soft material structure, or selective. Overall, the presentation is detailed in the way it is outlining the whole value chain of the process from the hardware exploited, the control of the hardware, details of solutions etc. It is thus considered to be of substantial interest to the soft materials community in general and DNA-hydrogel experts in more specific. 

We thank the reviewer for these remarks. 

2) The first of these is the apparent inhomogeneity of the fluorescence intensity within domains of the hydrogels – e.g. in red domains in Fig 6D. What is the reason for such inhomogeneity, and why does it appear more accentuated for the “red hydrogels” than for the other fluorophores?

We thank the reviewer for addressing this issue. Park et al. [Lab Chip 14, 1551–1563 (2014)] reported that various factors, including high UV exposure time and intensity, may result in non-uniform photopolymerization and crater-like fluorescence profiles. The red domain hydrogel in Figure 6D may have non-uniform polymerization and incorporation of the fluorescence DNA species; these non-uniformities may have become more visible after swelling. MADPH might be improved by optimizing the UV exposure time and intensity in order to achieve more uniform hydrogel photopolymerization. The results of our paper, however, still demonstrate that MAPDH could be used to pattern multi-domain DNA-crosslinked hydrogels, and that the different domains within this structure swell in response to different DNA swelling signals. 

We now discuss the possibility of non-uniform polymerization and how it might be improved in the Discussion section on page 21 line 517. We also added Park et al. [Lab Chip 14, 1551–1563 (2014)] as a reference when discussing this point (42).

42. Park S, Kim D, Ko SY, Park JO, Akella S, Xu B, et al. Controlling uniformity of asymerized microscopic hydrogels. Lab Chip. 2014 Apr 1;14(9):1551–63.

3) Please check figure numbering – there appear to be two figures nr 4, and figure 6 referred to in the text is understood as figure 5 of the manuscript (this is assumed in the above statement).

We thank the reviewer for pointing out this error. We fixed the numbering of the figures. Figures 5 and 6 in the revised version of the manuscript are now:

Fig. 5: Lift-off and collection of single-/multi-domain hydrogels using MAPDH-LC

Fig. 6: Swelling of collected single-/multi-domain hydrogels using MAPDH-LC

4) There appear to be some issue on presentation of references (e.g. Ref 4: Update year and add journal; Ref 9: There is an ACS Nano publication in 2020 with the same title – is that the one? Ref 10: missing year)

Thank you for pointing out these inconsistencies. We have corrected reference 4, reference 9, and reference 10 as follows:

[4] Scalise D, Rubanov M, Miller K, Potters L, Noble M, Schulman R. Programming the sequential release of DNA. ACS Synthetic Biology. 2020 Mar 26;9(4):749-55. 

[9] Yang S, Pieters PA, Joesaar A, Bögels BWA, Brouwers R, Myrgorodska I, et al. Light-activated signaling in DNA-encoded sender-receiver architectures. ACS Nano. 2020 Oct 20; 14(11):15992-16002.

[10] Jäkel AC, Heymann M, Simmel FC. Multiscale Biofabrication: Integrating Additive Manufacturing with DNA-Programmable Self-Assembly. Advanced Biology. 2023 Nov 3;7(3): 2200195.

Reviewer #2: 

1) The purpose of the manuscript is to describe a new platform to fabricate DNA-functionalized hydrogels. The topic is relevant and up to date, including the use of computational models to design and control the reproducibility of the printed patterns. 

We thank the reviewer for the comments above.

2) Although the authors are providing the full protocol as supplemental materials, a brief description of the methods should be included in the main document to facilitate understanding for the reader. 

We agree with the reviewer that a brief description of the methods should be included in the main text. We expanded the Methods section in the main text by providing details regarding MAPDH code, DNA sequences, hydrogel preparation, sacrificial layer removal, and DNA-crosslinked hydrogel swelling. Details of these changes are provided in our response to Reviewer 2’s point 6, which also discusses the protocol and its description.

3) Also, since there is no discussion section, the results should be discussed and compared with the literature in the results section. The results section presented in the current version is purely descriptive with no evident or very limited comparisons with the literature, and, most importantly, it lacks explanations about the meaning of each finding. The limitations of the study and the further applications of the method could also be better addressed at the end of the results section.

To address this concern, we have added a Discussion section to the manuscript to compare MAPDH to other methods more specifically and comprehensively than in the original version. 

In this comparison, MAPDH is contrasted with traditional hydrogel photopatterning methods. Conventionally, hydrogels with various functionalizations, including DNA, have been patterned using photomasks or replication molds [35][36]. These methods, while providing some degree of control, come with limitations in terms of predefined shapes and sizes, limiting their flexibility. Recent studies have attempted to overcome these limitations by integrating photopatterning tools, such as a digital micromirror device (DMD), with optical microscopes [17,37–39]. However, these approaches still involve manual ink injections, UV exposure, and manual positioning of the substrate, making them time-consuming and labor-intensive, especially for scaling up multi-domain patterning. Additionally, these methods often require significant ink volumes, raising concerns about cost efficiency, especially when dealing with high concentrations of modified DNA inks.

MAPDH addresses these challenges by automating ink flow, patterning, and washing, significantly reducing the need for manual interventions, and making the process less labor-intensive. Notably, MAPDH operates with much smaller ink volumes (100 µL), a substantial improvement compared to traditional methods that often involve milliliter volumes of inks. This reduction is crucial for cost efficiency, particularly when working with high concentrations of modified DNA strands. Furthermore, MAPDH's automated control minimizes ink wastage, allowing for precise application and reducing costs associated with ink usage. Future improvements focus on addressing current limitations, such as the low yield of the lift-off and collection process of multi-domain hydrogels, inhomogeneous photopatterning, and fluorescence. Ongoing work includes developing a robust sacrificial layer to anchor photopatterned hydrogels effectively, preventing misalignment after subsequent flow steps, and optimizing UV exposure time and intensity to achieve uniform photopolymerization.

Moreover, in the realm of biomedical applications, MAPDH's capabilities hold promise for various applications, including the design of controlled drug delivery systems [18], scaffolds [46], and organ-on-a-chip devices for tissue engineering [47]. These advancements underscore the potential of MAPDH in overcoming the limitations of traditional hydrogel photopatterning methods and establishing itself as a valuable tool in the biomedical field. We conclude this section with a description of potential applications of MAPDH to create hydrogels that demonstrate timed release or designed reactions, direct cell growth, or for local self-assembly. We believe MAPDH can also be useful for various biomedical applications, including designing drug delivery systems, tissue scaffolds, or organ-on-a-chip devices. 

In the Discussions section we also included the following references:

35. Huang F, Chen M, Zhou Z, Duan R, Xia F, Willner I. Spatiotemporal patterning of photoresponsive DNA-based hydrogels to tune local cell responses. Nat Commun. 2021 Apr 22;12(1):2364.

36. Simmons DW, Schuftan DR, Ramahdita G, Huebsch N. Hydrogel-Assisted Double Molding Enables Rapid Replication of Stereolithographic 3D Prints for Engineered Tissue Design. ACS Appl Mater Interfaces. 2023 May 31;15(21):25313–23.

42. Park S, Kim D, Ko SY, Park JO, Akella S, Xu B, et al. Controlling uniformity of photopolymerized microscopic hydrogels. Lab Chip. 2014 Apr 1;14(9):1551–63.

46. Liu AP, Appel EA, Ashby PD, Baker BM, Franco E, Gu L, et al. The living interface between synthetic biology and biomaterial design. Nat Mater. 2022 Mar 31;21(4):390–7. 

47. Liu H, Wang Y, Cui K, Guo Y, Zhang X, Qin J. Advances in Hydrogels in Organoids and Organs-on-a-Chip. Advanced Materials. 2019;31(50):1902042.

Additional comments:

4) The authors target functionalization with DNA, but the platform provides many more options than that. Those possibilities should be better explored in both the introduction and the discussion. You are proposing a method with various applications in biomedical engineering and tissue engineering. So, it could be interesting to reinforce that in your rationale.

We agree with the reviewer that our method could potentially be used for a variety of biomedical applications. To highlight these potential uses, we have included the following statement in the Discussions section (page 22 line 525):

Harnessing the biocompatibility and permeability of hydrogels, MAPDH may also be promising in various biomedical applications, such as designing controlled drug delivery systems [18], scaffolds [46], or organ-on-a-chip devices for tissue engineering applications [47]. 

We also added references [46] and [47] to the paper when adding this statement:

46. Liu AP, Appel EA, Ashby PD, Baker BM, Franco E, Gu L, et al. The living interface between synthetic biology and biomaterial design. Nat Mater. 2022 Mar 31;21(4):390–7. 

47. Liu H, Wang Y, Cui K, Guo Y, Zhang X, Qin J. Advances in Hydrogels in Organoids and Organs-on-a-Chip. Advanced Materials. 2019;31(50):1902042.

5) Define abbreviations upon their first appearance in the document.

To address the reviewer’s comment, we have provided definitions for all abbreviations in the revised version of the main text at the time these abbreviations are first used. These include Digital Micromirror Device (DMD; page 5 line 116), poly(ethylene glycol) diacrylate (PEGDA; page 6 line 135), Multi-domain Automated Photopatterning of DNA-functionalized Hydrogels (MAPDH; page 4 line 94), Polydimethylsiloxane (PDMS; page 10 line 222), and 1X Tris base, acetic acid, EDTA, 12.5 mM Mg2+ (1X TAEM; page 6 line 137). Definitions for the abbreviations of reagents used in this study can also be found in SI Section 33. 

6) Essential steps such as hydrogel preparation, DNA sequences, and functionalization should be included in the methods section. Although there is no need to describe the full code, it would be interesting to mention the language used for programming in the methods (Fig 1 and its description could be moved to the methods). Also, the swelling method and the sacrificial mold removal should be described in the main manuscript.

We appreciate the interest in providing these methodological details in the main text. We have expanded the Materials and methods section by providing details regarding the language used for the MAPDH code, DNA sequences and their functionalization, hydrogel preparation, sacrificial layer removal, and DNA-crosslinked hydrogel swelling.

7) Even though the authors claimed the method is simple, the complexity of developing microfluidic systems, the alternated crosslinking in microscale, and the need for postprocessing to remove the sacrificial molds are not simple. Therefore, the idea of the “simple method” should be interpreted with care.

We agree with the notion of what is simple is inherently subjective. As the reviewer states, for example, microfluidic methods may be complex for some investigators and more straightforward for others. We have removed the description “simple” and replaced this with more specific or quantitative features of the MAPDH method in the revised version of the manuscript.

---

## [Editor Report · Decision Letter 1]

4 Dec 2023

Multi-domain Automated Patterning of DNA-Functionalized Hydrogels

PONE-D-23-06068R1

Dear Dr. Schulman,

We’re pleased to inform you that your manuscript has been judged scientifically suitable for publication and will be formally accepted for publication once it meets all outstanding technical requirements.

Kind regards,

Nileshkumar Dubey

Academic Editor

PLOS ONE
---

## [Editor Report · Acceptance letter]

24 Jan 2024

PONE-D-23-06068R1 

PLOS ONE

Dear Dr. Schulman, 

I'm pleased to inform you that your manuscript has been deemed suitable for publication in PLOS ONE. Congratulations! Your manuscript is now being handed over to our production team.

Kind regards, 

on behalf of

Dr. Nileshkumar Dubey 

Academic Editor

PLOS ONE